# Injecting Sensitivity Constraint Into Continual Learning Significantly Enhances Surrogate-Aided Optimization

## Abstract

A myriad of scientific and engineering optimization and learning tasks involve running a numerical model to guide optimization directly or generate training data for function mapping algorithms. Surrogate models can greatly accelerate these tasks, but they often fail to capture the true input-output relationships (sensitivities) so they lose the ability to guide high-dimensional and long-horizon optimization. Online continual learning (OCL) – iteratively obtaining numerical results to continue training the surrogate – can mitigate this issue, but may still be insufficient. Here we propose scheduled injection of sensitivity constraints (SC, matching the Jacobian of the surrogate model with that of the true numerical model) for the surrogate into OCL to enforce realistic output-parameter relationships. We evaluate this approach across diverse datasets and optimization frameworks where continual surrogate training is used: (1) multi-objective multi-fidelity surrogate-assisted Bayesian optimization and Pareto front exploration; (2) hybrid end-to-end training of coupled neural networks and process-based models; and (3) a modified unifying framework for generative parameter inversion and surrogate training. Across all of these tasks, inserting SC accelerates the descent to optimality and consistently improves the main optimization outcome, as it critically improves the future trajectory of optimization. OCL improves data relevance and SC ensures sensitivity fidelity, and they together produces an efficient surrogate model that almost achieves the same effect as the full physical model, only achievable by OCL+SC. It consistently outperform pretrained-only surrogate models with SC or OCL without SC, not to mention the pretrained-only model without SC, so the benefits of two procedures reinforce each other. Even infrequent surrogate finetuning with SC injection (once every 5 epochs) can induce large benefits in optimization outcome. Together, these results demonstrate the possibility to enable large-scale optimization of complex systems for big-data learning and knowledge discovery.

## 1 Introduction

**Optimization with physical models:** Across various domains including robotic controls, physics, biomedicine, geosciences, etc., many crucial optimization tasks still involve physical models. The first is parameter estimation. Physical models solve governing equations and can ensure some expected behaviors, but they are often modulated by parameters that determine how the states evolve (Tarantola, 2004). Some of these parameters are physical parameters, i.e., thermal conductivity in heat transfer, or diffusion coefficients in porous media (Lu et al., 2021); Some of them are conceptual parameters to resolve the impacts of scale, i.e. parameterization schemes in climate modeling (Arakawa, 2004; Hourdin et al., 2017); Some others are empirical parameters to compensate for insufficient process representations (Kattge & Knorr, 2007; Medlyn et al., 2011). In many applications, the true values of these parameters or the parameterization function (the procedure to generate these parameters based on some known inputs) are not known in advance and must be inferred through iterative calibration or inverse modeling that ensuring a good match between simulations and observed system behaviors (Gupta et al., 1998). In more recent designs, we can train a neural

network to capture the inverse mapping from solution to the parameters (Ardizzone et al., 2019) from large amounts of accumulated simulations. For both, we need a large number of simulations.

The second optimization task gaining popularity is knowledge discovery, where a part of the physical model formulation is predefined while some parts of the relationships are to be learned using coupled neural networks (Karniadakis et al., 2021; Yang et al., 2017; de Avila Belbute-Peres et al., 2018). The physical components help narrow down what the neural networks (NNs) can learn, so we gain some interpretability of the learned relationship. Knowledge discovery often carries out end-to-end joint training of the hybrid system on big data. A third case is design optimization, where we select design variables to maximize certain benefits or explore the Pareto front for multiple objectives (Leifsson & Koziel, 2010; Wang & Shan, 2008; Praslicka et al., 2023). A fourth case is model predictive control (MPC) (Amos et al., 2019; Schwenzer et al., 2021), where a physical model represents the physical environment and its interaction with the body to be controlled. The algorithm then searches for optimal controls that achieve certain trajectories.

In all of the aforementioned optimization tasks, a process-based model is needed to represent the physical system behavior with some rigidity and it needs to be run a large number of times. The physical model may be solving partial differential equations (PDEs) or other types of equations, sometimes over long time periods or at high spatial resolution, which can be computationally expensive. The optimization can then be prohibitively expensive, especially when the parameter dimension is high.

**Surrogate-assisted optimization:** Surrogate models are designed to mimic the behaviors of the full model at a fraction of the cost, deliver significant computational savings and enable large-scale or real-time analyses. Gaussian process regression, neural networks and other machine learning methods have long been used as surrogate models (Marrel & Iooss, 2024). More recently, neural operators have gained attention due to their ability to learn mappings between infinite-dimension function spaces and propagate initial or boundary conditions through space and time, allowing them to predict solutions at arbitrary locations and timesteps rather than at fixed training points (Li et al., 2021; Wang et al., 2021; Azizzadenesheli et al., 2024). Yet, in the face of high-dimensional inputs, surrogate models are challenging to train and frequently lose fidelity when used outside their training knowledge domain or require an impractically large number of parameter samples to mimic the underlying system.

**Online continual learning:** The default strategy for surrogate models is to pretrain them with a diverse and comprehensive set of simulations so they accurately cover the entire input space during the optimization. However, this can be challenging with high-dimensional, long-horizon optimizations where it is difficult to devise the training data. As a result, the surrogate model may lose the ability to guide as the optimization proceeds. In contrast, online training strategies dynamically generate new training data during optimization and iteratively retrain the surrogate using fresh samples computed by the physical models (Meyer et al., 2023). This has already been a widely followed path in traditional surrogate-assisted optimization (Wistuba et al., 2018). Training solely the recent examples, however, risk catastrophic forgetting. This need for continual learning is not without some resemblance to training AI models under continuously-arriving new data (Aljundi et al., 2019; Nagabandi et al., 2019), or evolving agents for changing environments (Wołczyk et al., 2021; Zhou et al., 2024; Liu et al., 2025; Wu et al., 2021), where one needs to balance adaptation to new environments as well as mitigate catastrophic forgetting.

A more recent representative example in the machine learning era is the Self-directed Online Learning Optimization (SOLO) framework, as suggested by Deng et al. (2022), which integrates deep neural network (DNNs) as surrogates with finite element simulations to iteratively approximate objective functions across problems as Truss optimization and heat transfer enhancement. Similarly, continual learning approaches aim to mitigate forgetting by continuously updating the training buffer with representative samples. The buffer can be refreshed by replacing older samples chronologically, randomly, or through gradient-based selection strategies that prioritize data most informative for maintaining model accuracy (Aljundi et al., 2019). Here we will demonstrate that even online training may not adequately address the fidelity issue – it can still lead to suboptimal optimization outcomes which could be substantially enhanced by the incorporation of sensitivity information, when available, in surrogate training.

## 2 RELATED WORK: OPTIMIZATION FRAMEWORKS

We sample several successful optimization frameworks that utilize surrogate models and could benefit from our proposed idea of injecting sensitivity constraint.

**Multi-objective multi-fidelity Bayesian optimization (MOMF-BO):** (Takeno et al., 2020) is a class of methods that seeks to efficiently explore the Pareto front of multiple competing objectives under strict computational budgets. It combines evaluations of the true numerical model at different fidelity levels, e.g., using cheaper, coarser-resolution approximations alongside high-fidelity simulations — to balance accuracy with cost. The framework begins with a small initial set of evaluations to train a probabilistic surrogate model (typically a Gaussian process) that not only predicts objective values but also quantifies uncertainty. Based on these surrogate predictions at each step, an "acquisition function" selects promising candidates by balancing exploring uncertain regions and exploiting areas likely containing good solutions, dictated by a fidelity parameter (Irshad et al., 2024). These selected candidates are then evaluated using the actual numerical model at chosen fidelity levels, and the results are added to the buffer of ground-truth data. The surrogate is retrained on this expanding buffer, and the cycle repeats until the evaluation budget is exhausted to depict the Pareto fronts.

**Hybrid training of neural networks and differentiable numerical models:** A large class of methods train neural networks (NNs) together with physically-based numerical models that support differentiable programming (Innes et al., 2019). The physically-based component provides a behavioral backbone while the NNs learn either parameterization functions (Feng et al., 2022), or certain missing processes. To be trained with NNs jointly, the physics model is either re-implemented to enable automatic differentiation (AD) or solve the adjoint equations for gradients (Rackauckas et al., 2021; Gelbrecht et al., 2023), or a surrogate NN can be used in its place. An NN as the surrogate model can naturally support AD and joint training. In the end, the algorithm trains NNs to discover knowledge, provide physical parameters, and improve the predictive performance of the combined system.

**Joint training of generative parameter inversion and surrogate model:** Lingsch et al. (2024) proposed the Fast Unified Simulation and Estimation (FUSE) framework, designed to jointly address prediction tasks involving both discrete parameters and continuous fields governed by parametric PDEs. FUSE integrates a forward model and an inverse model into a single end-to-end differentiable setup. The forward model (implemented using neural operators) maps parameters and other inputs to the solution space over time. The inverse model, implemented using a Flow Matching Posterior Estimation (FMPE), generates the possible parameters satisfying the posterior distribution conditional on indirect or partial observations. The two models are trained in a combined system in an alternating order, with full model runs at the beginning to provide training data for the scheme. In the end, the training produces both a useful surrogate model and a parameter-generating model.

**Physics-informed or gradient-informed surrogate training:** To address the limitations of surrogate models, recent work such as physics-informed neural operators (PINO) (Li et al., 2024) has explored incorporating physical constraints into surrogate training. Towards incorporating physical constraint, Behroozi et al. (2024) proposed sensitivity-constrained Fourier Neural Operator (SC-FNO), incorporating a sensitivity-based term using the exact Jacobian of model output with respect to input parameters to regularize the gradient of the surrogate model. Their model outperformed standard and physical-equation-regularized FNOs for several parametric PDEs. Similarly encouraging results were proposed in Derivative-enhanced DeepONet (DE-DeepONet) (Qiu et al., 2024) and Derivative-informed Neural Operators (DINO) (O'Leary-Roseberry et al., 2024), which apply low-rank approximations to obtain approximate Jacobians, and Sobolev training (Czarnecki et al., 2017), which adds derivative terms to improve function approximation in Sobolev norms. Despite these advances, all such approaches have been demonstrated in offline or pretrained settings, typically with relatively low-dimensional inputs. It remains unclear whether these gradient-informed strategies remain effective in online continual learning, where surrogates are repeatedly updated and their evolving accuracy affects future optimization trajectories.

**Our contribution:** For each of the aforementioned optimization frameworks, we propose to incorporate a sensitivity constraint (SC) loss term, which matches the sensitivity of the surrogate model with that of the numerical model, while enabling online continual learning (OCL) of the surrogate model. We finetune the surrogate only upon scheduled intervals, e.g., once every epoch. We will demonstrate significant improvements of the main optimization from OCL+SC compared to OCL alone. Our contribution does not lie in inventing sensitivity-matching losses per se, but in the novel

coupling of OCL and SC and show how their combination, applied at various frequencies, influence long-horizon optimization. To our knowledge, no previous work showed how OCL interacts with SC. Unlike prior work that requires large-scale pretraining of gradient-informed surrogates, we show that iteratively finetuning yields superior efficiency, because the sensitivity information is used exactly where it matters—along the optimization trajectory and in regions critical to the search.

## 3 EXPERIMENTS

We evaluate our approach across three frameworks and four experiments/datasets. Each framework is sequential in nature; results in early iterations, as influenced by the surrogate models employed, determine future candidate designs or search directions. Therefore, we must assess not only the quality of the accuracy of the surrogate models but also the quality of the optimization outcome and the speed of convergence. By testing across a diverse selection of frameworks and datasets, ranging from low to high complexity, we motivate the general applicability of the continual training with sensitivity constraints (SC).

### 3.1 MULTI-OBJECTIVE MULTI-FIDELITY BAYESIAN OPTIMIZATION (MOMF-BO)

The Branin-Currin problem serves as a simple, analytical benchmark representative of typical MOMF-BO tasks. In particular, this benchmark pairs the Branin and Currin functions as two objectives (see Appendix A.1) with a known Pareto front (Irshad et al., 2024). In the HV-KG framework introduced by Daulton et al. (2023), differentiable Gaussian Process (GP) surrogate models for objective functions and an acquisition function leveraging knowledge gradients to achieve high efficiency and accuracy relative to other standard optimization search algorithms. Daulton et al. (2023) trained GP-based probabilistic surrogates by maximizing the log marginal likelihood (MLL) — a statistical objective that balances data fit and generalization. We augment this setup by supplementing MLL with a sensitivity-constrained loss term generally defined by

$$L_s = \frac{1}{M} \sum_{j=1}^{M} \left\| \frac{\partial \hat{\mathbf{u}}(\mathbf{x}_j, t_j; \mathbf{p})}{\partial \mathbf{p}} - \frac{\partial \mathbf{u}(\mathbf{x}_j, t_j; \mathbf{p})}{\partial \mathbf{p}} \right\|^2, \quad (1)$$

where $\mathbf{x}_j$ and $t_j$ represent spatiotemporal or otherwise design coordinates of points where Jacobians are evaluated, and $\mathbf{p}$ represents parameters that modulate the modeled system $\mathbf{u}$. This loss encourages the surrogate to learn accurate responses to dynamic parameter landscapes. For the Branin-Currin objective functions, this constraint is imposed on the fidelity parameter $s$ intended to simulate high and low-fidelity approximations of the true objectives (higher fidelity being more costly in practice). We then add surrogate model finetuning capabilities so that at every stage of optimization, each objective's surrogate is refined on a growing buffer of candidate optimal designs instead of being retrained from scratch. Algorithm 1 in the Appendix details our modifications to the HV-KG algorithm.

In order to evaluate the joint contribution of OCL and SC in this framework, we tested four schemes — all of which compute an initial buffer of random candidate designs, and grow the buffer by one newly acquired instance at every optimization iteration. **(i) Standard HV-KG:** The original framework where surrogates are retrained at every iteration with randomly initialized weights; **(ii) SC HV-KG:** incorporating an SC loss term alongside the MLL; **(iii) OCL HV-KG:** the surrogates are pretrained on the initial buffer and finetuned on the expanded buffer, inheriting weights from prior iterations; and **(iv) OCL+SC HV-KG:** combining finetuning with the SC-augmented loss.

Figure 3.1 shows the results of each experiment. OCL+SC leads to the best optimization outcomes (lowest hypervolume regret) and fastest convergence in the MOMF optimization (Figure 3.1). We observe that by finetuning the surrogates each optimization iteration instead of retraining from scratch (Standard HV-KG), both OCL and OCL+SC follow steeper regret curves that converge earlier, arriving at the lowest regret value achieved by HV-KG with only one-sixth the cost. By inheriting the weights from prior iteration, the surrogate model preserves local accuracy in promising design regions, enabling the surrogates to facilitate more accuracy candidate acquisition for the Pareto front. Incorporating SC (Offline w/ SC in 3.1) to align surrogate and ground truth sensitivities is also highly beneficial, as the model reaches nearly the lowest achieved regret, though the convergence rate is not as fast as OCL+SC. Showing each augmentation individually, the synergistic value

offered by both together is apparent in the OCL+SC curve, achieving both the lowest log regret and converging with the lowest cost.

Although this demonstration is on a low-dimensional synthetic task, the observed results indicate that continual learning and sensitivity regularization can work in concert, producing more sample- and cost-efficient optimizations. While there are costs associated with retraining the surrogate or the computation and injection of Jacobians for SC, such costs drastically improve the efficiency and outcome of the overall scheme. As dimensionality increases, we expect the benefits illustrated here to compound, a trend confirmed in the higher-dimensional experiments that follow.

## 3.2 HYBRID TRAINING OF NNS COUPLED TO PROCESS-BASED EQUATIONS

We evaluate our framework on two Earth system science tasks where neural networks (NNs) are trained jointly with process-based models. In both settings, an NN parametrizes the connected model, producing parameters for the process-based equation with supervision via loss between the physical model's simulation and observations. Case (A) is a hydrologic model for streamflow and flood prediction introduced by Feng et al. (2022); Case (B) is a plant hydraulics model simulating sap flow (Aboelyazeed et al., 2025). Both resemble systems of ordinary differential equations with long time integrations (730 time steps during training for Case A, 360 time steps for case B). The training loss is defined on streamflow (flow rate in rivers) for Case (A) and vegetation sap flow (movement of water through the tree's vascular system) for case (B). We consider two NNs: (i) the parameterization NN, the primary object of optimization, which maps static geoscientific covariates (e.g., terrain slope, soil texture) to physical parameters needed by the numerical models, and (ii) a surrogate NN, trained to approximate the full physical model. The surrogate NN takes dynamic drivers (precipitation, temperature, radiation), time-invariant geospatial attributes (soil texture, elevation), and physical parameters as inputs to return either streamflow or sap flow. In addition, gradients of the physical model simulations with respect to physical parameters were computed using PyTorch automatic differentiation for sensitivity-constrained training. To avoid the prohibitive cost of computing full-Jacobians across all time steps, we sampled a representative subset for gradient evaluation.

At a schedule time, we finetune the surrogate model based on a data loss between the output of the surrogate model and the full physical model, and optionally an SC loss for the gradient of the output with respect to the physical parameters. As the whole system is coupled end-to-end and trained using gradient descent, the gradient information through the surrogate model is part of the gradient chain to train the parameterization NN — we do not have direct observations for the parameters output by the NN. The algorithms and train-test splits are detailed in Appendix A.2. Each epoch contains approximately 194 NN parameter updates for the hydrological model, and 35 updates for the ecosystem model.

We incorporate online surrogate training and introduce a hyperparameter, $n_{SC}$, specifying the number of epochs after which the surrogate model is finetuned. Whenever the surrogate is pretrained or finetuned, SC is applied when that option is enabled. We compare multiple schemes. **(i) Offline:**

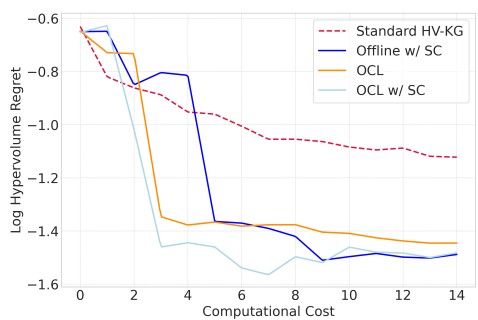

Figure 1: HV-KG log hypervolume regret versus cumulative analytical cost on the Branin-Currin MOMF benchmark. We compare the original HV-KG framework, HV-KG with online continual learning (OCL), HV-KG with sensitivity constraints (SC), and HV-KG with both OCL and SC. Log hypervolume regret quantifies optimization error and represents the gap between the hypervolume of the analytically derived true Pareto front and that of the identified solution set at a given cost. Standard HV-KG plateaus near regret $\approx -1.3$ at cost $\approx 51$, while OCL and OCL+SC converge faster and to lower regret. Note the cost axis only shows cost accrued during the acquisition of new candidate solutions (specifically, cost for evaluating the objective functions at different fidelities) up to the maximum cost budget and does not include the cost to assemble the initial surrogate training buffer (this initial cost is equal across all experiments here).

pretraining the surrogate on an initial buffer of physical model simulations and then used to replace the physical model during NN optimization, with its weights kept frozen; **(ii) OCL:** the surrogate model was pretrained and was subsequently finetuned (inheriting weights from previous training) once every $n_{SC}$ epoch on an iteratively refreshed buffer with new samples computed by the physical model; **(iii) OCL+SC:** adding loss into the surrogate training; **(iv) Offline+SC:** for comparison, we also ran with a surrogate model trained with a large initial buffer and sensitivity constraint. This scheme represents setup comparable to previous gradient-informed surrogate models like SC-FNO, DINO, DE-DeepONet or Sobolev training mentioned above. Furthermore, we include the high benchmark when the full **(v) physical:** model was used without surrogate.

### 3.2.1 HYDROLOGIC MODEL

Using the offline (pretrained) surrogate model with or without sensitivity constraint (SC) or the online trained model without SC (OCL) all cause large gaps in optimization metrics from the benchmark (training with the full physical model) in the hydrologic case (Table 1). The offline surrogate hydrologic model leads to the worst result, scoring an $R^2$ of 0.45 compared to the benchmark level of 0.72. $R^2$ of 0.45 is not an operationally acceptable ("poor") metric for flood forecasting while above 0.7 is considered good (Moriasi et al., 2015). This illustrates one of the key challenges of non-stationarity in this setting, as the region of interest to the parameterization NN in the parameter space is guaranteed to shift as the joint training proceeds. Incorporating either OCL or SC alone elevated $R^2$ to 0.63 and 0.65, respectively, which is a substantial gain. However, this gap is still inadequate. This means incorporating SC alone was useful,but the initial training buffer cannot satisfy the needs of optimization, especially for the high-dimensional surrogate. Incorporating OCL alone is also insufficient, which means adaptively casting points near the search optimum cannot provide adequate guidance, and even state-of-the-art adaptive learning strategies cannot produce a surrogate model of sufficient quality for this demanding task. It may be understood that the gradient of the surrogate model is part of the gradient chain for training the parameterization NN, and the learned surrogate sensitivities are often incorrect even if the prediction has high metrics.

Applying both OCL and SC nearly closes the gap of surrogate-assisted learning to the benchmark. By explicitly aligning surrogate sensitivities to those of the physical model, the surrogate achieves both high fidelity in predictive skill and stability in guiding parameter learning. This means the two options can be superimposed to leverage their compound benefits. Even with a low frequency of SC injection, $n_{SC} = 10$, the joint model performance ($R^2 = 0.68$ is noticeably higher than offline, OCL or offline+SC) (Table 1). At higher levels, the learning results are operationally sound. To completely reach the benchmark level, in the future, we envision a limited number of full-model iterations can be added toward the end to finish the optimization.

### 3.2.2 ECOSYSTEM MODEL

The ecosystem model joint training is also strongly aided by the use of OCL and SC, with improvements apparent for each procedure. The offline frozen surrogate model entirely failed the optimization, resulting in an $R^2$ of 0.21 and a large bias ($-0.059$) for the combined system, when the full model indicates the limit $R^2$ to be around 0.438 (see Table 1). The offline SC injection (offline+SC) elevated the performance up to an $R^2$ of 0.35, though it still lagged behind the benchmark. This is a prime example of pretrained surrogate model losing fidelity and guidance capability, and suggests that the system is too high dimensional and too complex to be captured by limited initial buffer. With online training only, $R^2$ increased significantly to 0.41 and the bias was reduced, underscoring the value of providing better fidelity near the optimality. The gap to benchmark level was further narrowed when employing online training with SC (OCL+SC). Injecting SC once per epoch ($n_{SC} = 1$), raised $R^2$ to 0.42, applying it every 2 epochs ($n_{SC} = 2$), pushed the performance further to an $R^2$ of 0.45, surpassing the benchmark.

### 3.2.3 COMMON OBSERVATIONS

Both experiments suggest OCL+SC can almost provide a model with the same effect as the full physical model even in high-dimensional case. The surrogate model needs to nearly perfectly reproduces the behaviors and sensitivities to reach this performance. The optimization constantly probes new locations around the present optimum and, with correct sensitivities from SC training, OCL+SC provides good guidance for new test points as ensured by Taylor series expansion. We

note that OCL+SC uses only $1/30$ of the computational cost in the ecosystem case (see timing in Table 5). Because of the complicated nonlinear solver and the needs to solve the adjoint equation for the gradients, the full ecosystem model is quite expensive – the optimization experiment using the full physical model requires 2 days on a V100 GPU. The surrogate model reduces an optimization run to almost 2 hours, so many learning experiments become plausible again. The costs of SC with $n_{SC} = 1$ is calculated as 37s per epoch or 26% of the time of OCL alone. Incorporating OCL and SC seem to be a necessary to enable a model like this for learning.

Both models show that merely infrequent injection of SC could already produce noticeable benefits. As we varied $n_{SC}$ from 1 to 10, as expected, we witness a change in performance. However, even with $n_{SC} = 10$, a gradient loss insertion once after hundreds of NN parameter updates, we still achieve a decently high performance (0.68 for the hydrologic model and 0.39 for the ecosystem model). Moreover, the computational cost is affordable. From OCL without SC to OCL+SC with $n_{SC} = 1$ raises the runtime per epoch by $\tilde{1}3\%$, not an extensive amount of cost to pay to obtain performance to raise $R^2$ from 0.63 to 0.71 for the hydrological model.

Table 1: Performance for the differentiable hydrological and ecosystem models with varying frequencies of gradient injection. Each differentiable model is trained to 50 epochs. Metrics including $R^2$ and bias are evaluated between the joint model outputs (streamflow or sap flow) and the observations using the full physical model and the trained parameterization NN. For online training with SC, we evaluate performance for continued surrogate training every 1, 2, 5, and 10 epochs. The time required to train the parameterization NN for one epoch in each case is given in seconds. Results for $R^2$ and Bias are reported as mean $\pm$ std for 5 random seeds.

| | Hydrologic Model | | Ecosystem Model | |
|---|---|---|---|---|
| | $R^2$ | Bias | $R^2$ | Bias |
| **Physical Model (Benchmark)** | 0.72 | 0.605 | 0.438 | -0.011 |
| **Offline** | $0.45 \pm 0.07$ | $0.70 \pm 0.05$ | $0.21 \pm 0.12$ | $-0.059 \pm 0.012$ |
| **OCL** | $0.63 \pm 0.03$ | $0.69 \pm 0.05$ | $0.41 \pm 0.02$ | $-0.022 \pm 0.005$ |
| **Offline + SC** | $0.65 \pm 0.05$ | $0.69 \pm 0.05$ | $0.35 \pm 0.04$ | $-0.030 \pm 0.015$ |
| **OCL + SC (1)** | $0.69 \pm 0.02$ | $0.64 \pm 0.02$ | $0.42 \pm 0.03$ | $-0.019 \pm 0.005$ |
| **OCL + SC (2)** | $0.71 \pm 0.02$ | $0.62 \pm 0.02$ | $0.45 \pm 0.02$ | $-0.014 \pm 0.003$ |
| **OCL + SC (5)** | $0.70 \pm 0.03$ | $0.63 \pm 0.02$ | $0.43 \pm 0.02$ | $-0.017 \pm 0.003$ |
| **OCL + SC (10)** | $0.68 \pm 0.04$ | $0.67 \pm 0.03$ | $0.39 \pm 0.06$ | $-0.023 \pm 0.005$ |

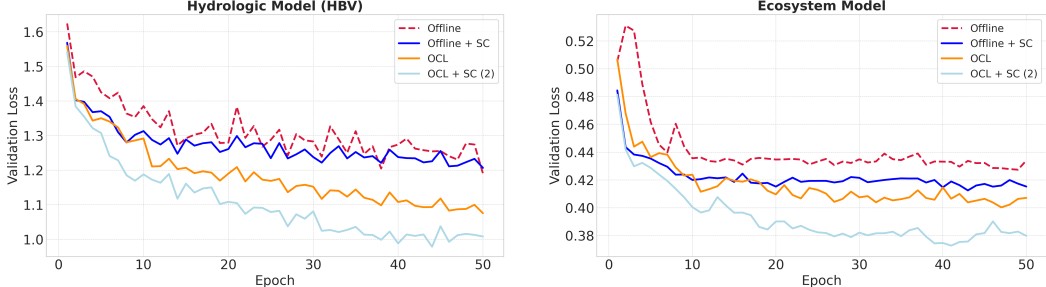

Figure 2: Validation loss during differentiable model training when using a surrogate pretrained offline with or without SC, and in an OCL scheme with or without SC. Results for the hydrological and ecosystem models are given in the left and right panels, respectively.

### 3.3 FUSE (JOINT GENERATIVE INVERSION-SIMULATION TRAINING)

We implemented the original FUSE design (Lingsch et al., 2024) while introducing the following two modifications to enhance its performance: 1) online FMPE training using data generated by the surrogate: the forward surrogate is used to generate new samples during training to be added to the buffer and boost the training data size for the FMPE inverse model; 2) using a sensitivity-constrained

FNO for the surrogate model (Behroozi et al., 2024) (details in Appendix A.3). For all configurations we tested, the full physical model was called only upon initial training data preparation, not iteratively during training. We evaluated these configurations on a benchmark 2D steady-state Darcy flow equation with Dirichlet boundary conditions, used in earlier FNO research (Li et al., 2024). Following their approach, we generated 1000 paired samples of $(a,u)$ on a 32 x 32 spatial grid to train and evaluate different cases, where $a(x, y)$ represents the spatially varying diffusion coefficient and $u(x, y)$ represents the solution field. In FUSE setup with SC injected, the FNO is trained to predict the solution field $u$ from the diffusion coefficient $a$ using a combined loss function $(L_{\text{data}} + \lambda L_{\text{SC}})$. Where $L_{\text{data}}$ measures the discrepancy between the predicted and true-u fields, $L_{\text{SC}}$ penalizes mismatches between predicted and true sensitivities $\partial u / \partial a$, and the weighting factor $\lambda$ is set to 1. The FMPE model is trained in parallel to approximate the posterior distribution $\rho(u|a)$ given spatially masked observations of $u$, using Fourier-projected representations of $u$ as conditional inputs.

Table 2: Results for FUSE applied to Darcy flow (steady state) with a 25% spatial mask applied to the $u$ field input to the FMPE. Continuous Ranked Probability Score (CRPS) quantifies how closely the generative FMPE's parameter estimates matches the true $a$ field, while Relative $L_1$ and $L_2$ are the error norms of the surrogate-simulated $u$ field. Rows labeled **True** $a$ report forward surrogate evaluation using ground-truth diffusion coefficients, while rows labeled **Predicted** $a$ reports the full FUSE pipeline evaluation, where FMPE first predicts $a$, and the surrogate then maps $a \to u$. The difference between the two quantifies the impact of $a$ generation error on the forward simulation $u$.

| Model | Experiment Setup | CRPS | Rel. $L_1$ Error | Rel. $L_2$ Error |
|---|---|---|---|---|
| **FUSE** | True $a$ | — | $2.91 \pm 0.86$ | $3.71 \pm 0.89$ |
| | Predicted $a$ | $0.72 \pm 0.16$ | $8.40 \pm 2.94$ | $9.72 \pm 2.83$ |
| **Online FUSE** | True $a$ | — | $2.95 \pm 0.88$ | $3.76 \pm 0.90$ |
| | Predicted $a$ | $0.74 \pm 0.17$ | $9.64 \pm 3.48$ | $10.84 \pm 3.30$ |
| **Online FUSE + SC** | True $a$ | — | $2.59 \pm 0.84$ | $3.28 \pm 0.83$ |
| | Predicted $a$ | $0.48 \pm 0.15$ | $5.65 \pm 2.63$ | $6.50 \pm 2.48$ |

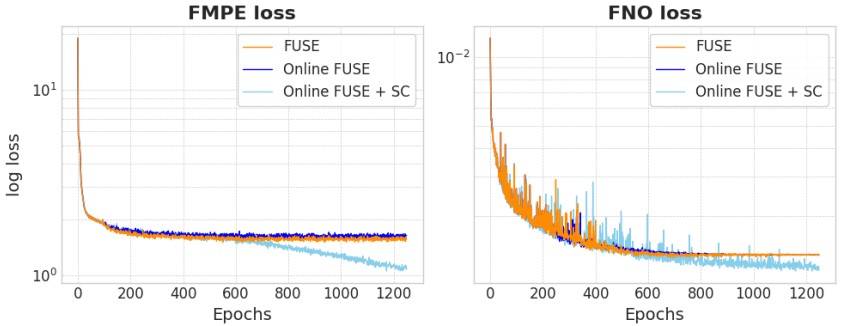

Figure 3: Validation loss during joint training for three configurations; FUSE, Online FUSE, and Online FUSE+SC.

OCL+SC shows clear advantage for the FMPE training as well as the surrogate training over standard FUSE (Table 2 and Figure 3). The FUSE and Online FUSE (OCL without SC) cases produced very similar results, and the latter has even slightly larger errors both for the FMPE (CRPS) and the forward simulation ($L_1$ and $L_2$ errors) (see the first two rows for each configuration in Table 2 and the orange/blue lines in Figure 2). This means using a surrogate model that is trained without SC does not produce simulations of high enough quality to inform FMPE. In contrast, Online Training w/ SC (Online FUSE+SC) lowered the CRPS to 0.478, $L_1$ to 5.65 and $L_2$ to 6.50. All of them are around 34% lower than either standard FUSE or Online FUSE without SC. The OCL+SC continues to push down FMPE error around 600 epochs, when the offline or the OCL without SC models stall (Figure 3). Note here the new training data for FMPE is provided by the surrogate model only and not additional full physical model runs. This means the generative FMPE can indeed benefit from a

higher-quality surrogate model trained on both data and the sensitivities. Even if the surrogate model loss appears only slightly better (Figure 3 Right), its better internal relationships and robustness to perturbations (Behroozi et al., 2024) allowed it to generate more accurate new examples to improve FMPE. In a positive feedback, the more accurately generated $a$ field, in turn, substantially lowered the forward $L_1$ and $L_2$ errors.

### 3.4 FURTHER DISCUSSION AND CONCLUSION

We have tested Online Continual Learning with Sensitivity Constraint (OCL+SC) across four different datasets and three algorithmically distinct classes of long-horizon optimization frameworks that use surrogate models in different ways: multifidelity Pareto-front exploration, hybrid training, learning inverse maps. None of these frameworks (or many other for this matter) previously employed SC. Nor did any of the related studies comment on algorithms similar to SC as an option for substantial improvements. Diverse strategies in active learning (Ren et al., 2021) work similarly as OCL and the acquisition functions in the MOMF-BO case, but no previous effort employed SC. Yet, in each case, the incorporation of SC for surrogate training produced immediate and unmistakable improvements, with OCL+SC approaching an equivalent tool as the full physical model with orders-of-magnitude lower cost. Therefore, we argue that the incorporation of SC represents an underexplored opportunity to usher in widespread performance gains and practical utility across scientific and engineering domains. Considering recent advances in hyper-efficient AI-enabled high-resolution solutions to PDEs, e.g., neural operators, and the need of AI agents for world models, the applicable scenarios of SC in complex optimization tasks should only grow rapidly.

While sensitivity-based regularizers themselves have been proposed before, our work is the first to demonstrate their effectiveness when embedded in online continual surrogate learning loops. In this regime, surrogate accuracy is not only about pointwise fit but directly influences the future trajectory of the optimization. It was not obvious that sensitivity constraints and OCL would reinforce each other. Our ablations on different SC frequencies confirm that the combination is synergistic, creating a new design space for long-horizon surrogate-aided optimization. To our knowledge, no prior work has established this. As compared to earlier work using pretrained sensitivity-aware surrogate models (Behroozi et al., 2024; Qiu et al., 2024; O'Leary-Roseberry et al., 2024), the usefulness of OCL algorithms means we do not need to collect comprehensive and exhaustive pretraining data for the surrogate, which can be implausible for high-dimensional and long-horizon optimization task. Instead, we can spend limited computational resources along the main optimization in a more targeted fashion. The fact that the SC adds strong benefits beyond OCL alone liberates AI surrogates for many high-resolution, high-computational-demand applications.

Previous studies of surrogate models primarily focused on efficiency and forward predictions, but have severely underestimated the importance of the correct internal relationships (quantified by sensitivities), generalizability against perturbations, and their accumulative effects. In agreement with Cao et al. (2025), we argue this is a primary weakness with the majority of present surrogate models. In fact, as we searched the literature, while neural operators are widely used for forward prediction, their use within inverse algorithms (not directly learning inverse maps) is still relatively rare. The purpose of a model in an optimization is to explore, to perturb, and to guide. If we use surrogate model to generate training data the inverse mapping as in the FUSE case above, it clearly also requires the model to be error-resistant when the inputs are perturbed. The fact that seemingly high-scoring surrogate model can do poorly in these tasks is not emphasized enough in the community. The tested frameworks have a long optimization horizon. Although at a given (especially initial) step, the direct prediction error may be very low, the accumulative and sensitivity error mean that using the surrogate trained on data alone for guidance is futile.

A critical element for newly enabled SC is the rapid computation of gradients from numerical models. Differentiable programming can support the calculation of high dimensional Jacobian, but finite difference, adjoint-based methods (Behroozi et al., 2024), and low-rank approximations (Qiu et al., 2024) can also be used. As differentiable programming is gaining popularity in many domains (Ramsundar et al., 2021; Gelbrecht et al., 2023; Shen et al., 2023; AlQuraishi & Sorger, 2021; Zhu et al., 2024), we expect more and more numerical models to directly support it.

REPRODUCIBILITY STATEMENT

The Branin–Currin HV-KG experiments build on publicly available code from `https://github.com/meta-pytorch/botorch`, and the FUSE experiments build on `https://github.com/camlab-ethz/FUSE`. Both repositories can be adapted with the modifications described in this paper to reproduce the reported OCL and SC results. The differentiable hydrologic and ecosystem model experiments rely on code that is currently under internal development; this code will be released publicly upon completion of the ongoing open-sourcing process to enable full replication of these results.

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

# A  APPENDIX

## A.1  MULTIOBJECTIVE MULTIFIDELITY EXPLORATION OF THE PARETO FRONT

Daulton et al. (2023) deploy a hypervolume knowledge gradient (HV-KG) multi-objective optimization framework. As proposed in their study, we apply our online paradigm to the classical Branin-Currin synthetic multi-objective multi-fidelity (MOMF) optimization problem. This synthetic problem is characterized by the joint optimization of the Branin

$$B(\boldsymbol{x}) = a(x_2 - bx_1^2 + cx_1 - r)^2 + p(1 - t)\cos(x_1) + p,  \tag{2}$$

and Currin

$$C(\boldsymbol{x}) = \left[1 - \exp\left(-\frac{1}{2x_2}\right)\right] \frac{2300x_1^3 + 1900x_1^2 + 2092x_1 + 60}{100x_1^3 + 500x_1^2 + 4x_1 + 20}  \tag{3}$$

objective functions, where $x_{11} = 15(x_1) - 5$, $x_{22} = 15x_2$, $a = 1$, $b(s) = 5.1/(4\pi^2) - 0.01(1-s)$, $c(s) = 5/\pi - 0.1(1-s)$, $r = 6$, $p = 10$, and $t(s) = (1/(8\pi)) + 0.05(1-s)$ with fidelity parameter $s \in \mathbb{R}$ (Irshad et al., 2024).

---

**Algorithm 1** Sensitivity-Constrained MOMF optimization

---

1: **Input:** Initial computational budget $C_{\text{initial}}$, maximum evaluation budget $C_{\text{max}}$
2: Initialize buffer $B$ with samples evaluated under cost $C_{\text{initial}}$
3: Pre-train surrogate models on buffer $B$ *
4: **while** $C \leq C_{\text{max}}$ **do**
5:      **if** not 1st iter **then**
6:          Train surrogate models on buffer $B$ *
7:      **end if**
8:      Acquire new candidate points via acquisition function
9:      Evaluate candidates at chosen fidelities and append to buffer $B$
10:      Add cost for new candidates to cumulative cost
11: **end while**
12: Train final surrogate models on buffer $B$
13: Compute optimal Pareto frontier

---

* Denotes procedures for online learning with SC.

---

## A.2  HYBRID TRAINING OF COUPLING NEURAL NETWORKS AND PROCESS-BASED EQUATIONS

We used the physical models to generate training buffers for the surrogates including dynamic input data with dimensions *batchsize × timesteps × number of dynamic variables*, static attributes with dimensions *batchsize × number of static variables*, physical model parameters with dimensions *batchsize × timesteps × number of parameters*, corresponding physical model outputs with dimensions *batchsize × timesteps × number of outputs*, and Jacobian sensitivities with dimensions *number of outputs × batchsize × timesteps × number of parameters*.

**Hydrologic model:** For hydrologic (streamflow) applications, we use the differentiable model dHBV first introduced by Feng et al. (2022)) which couples an LSTM NN with the conceptual hydrologic model Hydrologiska Byråns Vattenbalansavdelning (HBV). HBV is a collection of differentiable bucket-based physical processes that are parameterized by 12 time-invariant, spatially-distinct parameters learned by the NN.

Following the standard of approach for this model demonstrated by Feng et al. (2022), we use the open-access Catchment Attributes and Meteorology for Large-sample Studies - Dataset (CAMELS) produced by Newman et al. (2022), which provides access to all dynamic inputs, attributes, and the target streamflow observations to use as ground truth. This dataset includes 34 years of data at a selection of 671 USGS gages around the continental US. Keeping with the standard benchmarks for this model in our experiments, we select a subset of 531 of these gages and and use train data from 1 October 1999 to 30 September 2008, and validation data 1 October 1989 to 30 September 1999.

For this model, we chose to implement an LSTM surrogate for its ability to handle the highly dynamic nature streamflow data, which can oscillate between high and low flow extremes in response

---

**Algorithm 2** Sensitivity-Constrained Training for ML-physics hybrid models

---

1: **Input:** Initial buffer $B$, maximum epochs $E$
2: Initialize buffer $B$ with batched physical model simulations
3: Pre-train surrogate model on buffer $B$ to resemble the physical model outputs *
4: **for** epoch = 1 **to** $E$ **do**
5:     Train NNs using data loss between surrogate model simulations and ground-truth observations
6:     Perform full model evaluation on random batches using the physical model
7:     Update buffer $B$ with newly generated samples *
8:     Update surrogate model by backpropagating from combined loss:
        $L_{\text{combined}} = L_{\text{data}} + \lambda L_{\text{SC}}$ (SC loss for selected parameters) *
9: **end for**
10: Evaluate full model using physical model without surrogate

---

* Denotes procedures for online learning with SC.

---

to the influx of water during precipitation events. We made the initial training buffer for this surrogate equivalent in spatio-temporal scope of training data as seen by the differentiable model's parameterization LSTM during training (531 locations and 3288 timesteps, as mentioned above). In particular the buffer contains 194 minibatches of data that are each seen once per training epoch. These minibatches include the dynamic drivers with shape $[100, 730, 3]$, static attributes with shape $[100, 35]$, parameters with shape $[100, 730, 192]$, and streamflow model output and ground truth, both with shape $[100, 730, 1]$. Jacobian sensitivities of the output to the input parameters are also stored with shape $[1600, 2, 4]$, where we stored the four most sensitive parameters for the middle and last timesteps of each minibatch optimization window. We try generating parameters for the buffer with both Latin Hypercube Sampling (LHS) to uniformly sample the entire parameter space, as well as the parameterization LSTM itself (trained for 1 epoch).

Table 3: Descriptions for HBV model parameters.

| Parameter | Description |
|---|---|
| $\beta$ | Recharge non-linearity (-) |
| FC | Field capacity (mm) |
| $K_0$ | Runoff coefficient (day$^{-1}$) |
| $K_1$ | Subsurface flow coefficient (day$^{-1}$) |
| $K_2$ | Groundwater storage coefficient (day$^{-1}$) |
| LP | Wilting point as fraction of FC (-) |
| c | Percolation rate (mm·day$^{-1}$) |
| UZL | Upper zone storage limit (mm) |
| TT | Threshold temperature for snowfall (°C) |
| CFMAX | Degree-day factor (mm · °C$^{-1}$ · day$^{-1}$) |
| CFR | Refreezing coefficient (-) |
| CWH | Water holding capacity as a fraction of current snowpack (-) |

**Ecosystem model:** We used the developed differentiable ecosystem model ($\delta_{psn}$) described in Aboelyazeed et al. (2025) whose physical component is based on the photosynthesis and plant hydraulic modules originally derived from the Functionally Assembled Terrestrial Ecosystem Simulator (FATES Development Team, 2020). The parameter calibration module is composed of three multi-layer perceptron NNs which map static and dynamic inputs to 7 physical parameters required by the physical component in $\delta_{psn}$ (see Table 4). Among these parameters, $V_{c,max25}$ is dynamic, while the rest of the parameters are static.

For this model, we chose to implement an FNO surrogate due to its high computational efficiency which is highly required for such expensive physical model. For the initial buffer, we ran the differentiable framework (physical model + parameterization NNs) for one epoch, during which the parameterizations NNs were trained concurrently with data generation. The initial buffer is composed of 35 batches, each with a batch size of 100 trees. Each batch includes: (i) dynamic inputs of size $[100, 360, 14]$, (ii) static attributes $[100, 4]$, (iii) physical parameters $[100, 360, 70]$, (iv)

sap flow simulations [100, 360,1], and (iv) sensitivity gradients [100, 360,7]. Here 100 refers to the batch size, 360 to the number of timesteps, 14 to the number of dynamic inputs, 4 to the static attributes, 70 to the physical parameters (of which 7 are learnt by NNs). For the sensitivity gradients, we computed the derivative of the sap flow simulation at the last timestep with respect to the learnable parameters.

For observation data, we used Tree-level sap flow observations from SAPFLUXNET database (Poyatos et al., 2021), which compiles hourly records from sites around the world. For each site, we selected three-month periods during growing season at hourly resolution, using two months for training and one month for testing. Our selected dataset includes observations from more than 100 sites and covers more than 120 species across different plant types from only overstory trees that receive full-sunlight. Beside sap flow observations, we used weather data provided by SAPLUXNET (i.e., temperature, radiation, wind speed, relative humidity), and tree-specific attributes such as tree height, diameter at breast height, and sapwood area. We further incorporated hourly soil moisture from ERA5 Reanalysis dataset (Muñoz Sabater, 2019), as well as soil texture data (sand percentage, clay percentage, organic matter) from soilGrids-ISRIC (Poggio et al., 2021) at multiple depths. Species-specific traits such as wood density and specific leaf area were obtained from TRY database (Kattge, 2020), while hydraulic traits were computed as described in (Christoffersen et al., 2016).

Table 4: Descriptions for Ecosystem model parameters.

| Parameter | Description |
|---|---|
| $V_{c,\max25}$ | Maximum carboxylation rate at 25 °C |
| $P_{50,gs}$ | Water potential at 50% loss of conductivity for stomata |
| $a_{gs}$ | Shape parameter for stomatal vulnerability |
| $g_1$ | Medlyn slope parameter for Medlyn stomatal conductance model (Medlyn et al., 2011) |
| $K_{s,\max,x}$ | Maximum xylem conductivity per unit sapwood area |
| Leaf scaler coeff$_1$ | Coefficient one for decrease in leaf Nitrogen through the canopy (Lloyd et al., 2010) |
| Leaf scaler coeff$_2$ | Coefficient two for decrease in leaf through the canopy (Lloyd et al., 2010) |

Table 5: Average runtime per epoch for the differentiable ecosystem model. All configurations were trained for 50 epochs. For online training with sensitivity constraint (SC), we evaluate performance for continued surrogate training every $n_{SC} =$1, 2, 5, and 10 epochs. The numbers in the parentheses is $n_{SC}$. Reported times in seconds represent the full simulation time divided by the total number of epochs. Because OCL finetunes the surrogate for each epoch, it can be more expensive than OCL+SC(5) and OCL+SC(10) even if it does not compute the Jacobians. At $n_{SC} = 1$, the cost of computing Jacobians and training on SC is calculated as $180s - 143s = 37s$ per epoch, or 26% of the time of OCL alone.

| | Runtime per Epoch (s) |
|---|---|
| **Physical Model(Benchmark)** | 4176 |
| **Offline** | 17 |
| **OCL** | 143 |
| **Offline + SC** | 21 |
| **OCL + SC (1)** | 180 |
| **OCL + SC (2)** | 144 |
| **OCL + SC (5)** | 120 |
| **OCL + SC (10)** | 107 |

## A.3 FUSE (JOINT GENERATIVE INVERSION-SIMULATION TRAINING)

---

**Algorithm 3** Sensitivity-Constrained FUSE

---

1: **Input:** Initial buffer $B$, maximum epochs $E$, warm-up epochs $E_w$
2: Initialize buffer $B$ with batched data
3: **for** epoch = 1 **to** $E$ **do**
4:     Train FMPE inverse model using $\mathbf{L}_{\text{FMPE}}$
5:     Train FNO forward model using
        $L = L_{\text{data}}$ (data loss only) or $L = L_{\text{data}} + \lambda L_{\text{SC}}$ (with sensitivity-constrained loss) *
6:     **if** epoch $\geq E_w$ **then**
7:         Generate additional training samples using FNO surrogate *
8:         Update buffer $B$ with generated samples *
9:     **end if**
10: **end for**
11: **Evaluate:**
        (a) Combined SC-FUSE model (FMPE + FNO) for full-inversion prediction task, or *
        (b) FNO forward model using ground-truth parameters
* Denotes procedures for online learning with SC.

---

