# OpenReview forum: "Injecting Sensitivity Constraint Into Continual Learning Significantly Enhances Surrogate-Aided Optimization"
_ICLR.cc/2026/Conference — ICLR 2026 Conference Withdrawn Submission_

### Official Review · Reviewer_73Xu · 2025-10-15

**Soundness:** 2
**Presentation:** 1
**Contribution:** 2
**Rating:** 2
**Confidence:** 4

**Summary:**

The authors study how to introduce Jacobian matching between surrogate and oracle functions into online continual learning - specifically in how to improve the surrogate model by matching not only the function values, but also their derivatives. The method is explored across 4 different tasks, including a combination of both synthetic benchmarks and real-world environments.

**Strengths:**

1. The contribution of this work is interesting and intuitive.
2. I appreciate the ablation over the frequency of SC injections, and also the experiments on how the method affects the overall runtime.

**Weaknesses:**

3. The sensitivity loss weight hyperparameter lambda seems to be manually tuned per task. Since gradient magnitudes differ dramatically across models and domains, performance may be sensitive to lambda. There is no relevant ablation study or discussion of this limitation from what I can see.
4. It is challenging for me to understand the Introduction within the context of the larger scope of this work, and required multiple reads at least on my end. For instance, what was the point of discussing knowledge discovery in the second paragraph? Catastrophic forgetting is also not explored at all in this work.
5. The sensitivity loss term necessitates the ability to compute (or at least approximate) first order derivatives of the true numerical model. This is not necessarily the case in practice (e.g., discrete design spaces, wet-lab experiments, MD simulators, etc).
6. $n = 5$ seeds is too few to make any meaningful conclusions – for example, I believe the 95% CIs for Offline + SC and OCL + SC (2) would overlap for the $R^2$ metric for the Hydrologic Model in Table 1.
7. There doesn’t seem to be actual experimental evidence of catastrophic forgetting in the paper. Did the authors observe any signs of catastrophic forgetting or degradation in performance outside the immediate optimization regions (e.g., checking overall $L_2$ error on the entire initial buffer B) in the OCL-only versus OCL+SC cases?
8. The manuscript would generally benefit from additional proofreading and editing. I generally do not feel super strongly about having a submission being 100% perfect in terms of grammar and spelling, but there are a large number of grammatical mistakes in this work that made it challenging to read through and required a couple of iterations to understand. I started detailing a few below in the "Minor Comments" section, but stopped somewhere in the middle so the list is by no means exhaustive.
9. I understand that the SC loss term is not the main contribution, but rather empirically studying how it can be effectively incorporated into OCL in different settings. However, this was only studied in only 3-4 applications, and the ablation studies are significantly lacking (e.g., see point 1 above, also ablating buffer size, number of warm-up epochs, dataset sizes, Jacobian approximation accuracy, etc). This makes my enthusiasm for the empirical contribution of this work significantly tempered.
10. I would also consider adding experience replay, elastic weight consolidation, and gradient episodic memory as baseline methods to compare against.

Minor Comments:
 - Abstract: "they together produces" should be "they together produce"
 - Abstract: "an efficient surrogate model" - I don’t think the authors mean "efficient" here, should it be "effective"?
 - Abstract: "consistently outperform" in line 31 should be "consistently outperforms"
 - Line 68: I’m not quite sure what "rigidity" means here.
 - Line 77: "infinite-dimension" should be "infinite-dimensional"
 - Line 90: "solely the recent examples" should be "solely on the recent examples"
 - Line 181: I don’t think HV-KG is a well-known acronym – the authors should define it first.
 - Line 181: The sentence that begins with "In the HV-KG framework" is not a complete sentence.
 - In general, it is unclear when better metric values are higher vs lower. Up and down arrows should be added to clarify.

**Questions:**

11. What is the definition of $M$ in equation (1)?
12. How is the value of lambda in Algorithms 2 and 3 determined?
13. In line 255, the authors state that a subset of the gradients were used to estimate the full Jacobian? What exactly was this algorithm?
14. Why is the performance of OCL + SC (2) better than OCL + SC (1) in Table 1? Shouldn’t more frequent continued training improve the performance?

---

> ### Author Response · Authors · 2025-11-26
>
> We are grateful for the reviewers’ careful and constructive comments on our submission. However, we note that the numerical scores appear more critical than the written assessments might suggest — which were largely encouraging and requested reasonable refinements. While we would address aforementioned concerns in a revised manuscript, we wished to first note this potential misalignment for the reviewers consideration.
>
> 1. What is the definition of M in equation (1)?
>
> In equation (1), M denotes the number of samples in the training dataset, used here to get the mean loss across all samples.
>
> 2. How is the value of lambda in Algorithms 2 and 3 determined?
>
> To ensure balanced weighting across the Ldata and LSC, we employed an automatic loss weighting scheme that learns a trainable parameter λ to balance each term in the loss function (including data loss, and SC loss for each parameter), which is learned concurrently during surrogate model training.
>
> 3. In line 255, the authors state that a subset of the gradients were used to estimate the full Jacobian? What exactly was this algorithm?
>
> Thank you for this question. Both cases (A) the hydrology model and (B) the ecosystem model evolve forward in time and can have time-dynamic inputs. Because both models are also recurrent in time, the output timeseries can depend on the entire timeseries of learned parameter inputs. Therefore, the full Jacobian representing the complete set of physical parameter sensitivities corresponds to the derivatives of the model outputs at all timesteps with respect to all parameters. To take a “subset” of the gradients, we refer to evaluating gradients at a portion of the target’s timesteps; dyt /dP. For example, this could look like using only the final timestep, or using the final timestep and an intermediate timestep, rather than the full temporal sequence in the target, y. This is necessary for computational efficiency, as creating the initial buffer dataset and training on all timesteps can both be very costly. There are more advanced ways to implement this selection (e.g., randomizing Jacobians used per batch and per training iteration), however, we have not fully evaluated these in the interest of obtaining a general consensus on the interaction of OCL + SC. We will elaborate on the rationale for this choice in our revisions.
>
> 4. Why is the performance of OCL + SC (2) better than OCL + SC (1) in Table 1? Shouldn’t more frequent continued training improve the performance?
>
> This is directly related to different hyperparameter settings used in training the models listed in Table 1. These settings are the number of epochs used to: 1) train the surrogate model at initialization, and 2) fine-tune during parameter inversion. Both  OCL + SC (1) and OCL + SC (2) use the same number of pretraining epochs, but they differ in how frequently the surrogate is fine-tuned. In OCL + SC (1), the more frequent fine-tuning may cause mild overfitting, whereas OCL + SC (2) represents a more relaxed update schedule that appears to generalize better. We hypothesize that if the surrogates were pretrained for fewer epochs at initialization, more frequent fine-tuning may yield improved performance.

---

### Official Review · Reviewer_p5Sm · 2025-11-01

**Soundness:** 2
**Presentation:** 2
**Contribution:** 3
**Rating:** 2
**Confidence:** 3

**Summary:**

The paper addresses the challenge of reduced guidance capability in high-dimensional,
long-horizon optimization. By integrating sensitivity constraints (SC) with online continual learning (OCL),
the authors present a generalizable framework that can be applied across diverse modeling contexts.
The paper demonstrated their effectiveness of this approach in several domains, including multi-objective multi fidelity
Bayesian optimization, hybrid training of coupled neural networks and process-based models, and generative parameter inversion with surrogate training,
highlighting its versatility and potential broad impact.

**Strengths:**

1. The OCL+SC framework is versatile and can serve as a plug-in for multiple models,
potentially broadening its impact across various applications.
2. The paper provides a clear and detailed explanation of the experimental settings, including application in multiple tasks

**Weaknesses:**

1. The paper does not clearly specify which Online Continual Learning (OCL) algorithm or variant
is employed among the many existing approaches (e.g., RAR[1], OCA[2],...)

2. The design of the sensitivity constraint is insufficiently justified.
There are multiple ways to inject or formulate sensitivity information
(e.g., input–output Jacobian norms, local Lipschitz bounds), yet the paper adopts one specific form without discussion.
It would be valuable to clarify why this particular sensitivity formulation was chosen,
how it affects optimization behavior, and whether alternative forms were tested.

3. Experimental comparison is limited; important baselines such as MF-OSEMO [3] and iMOCA [4] are missing in MO-MFBO.
The evaluation on MO-MFBO were mainly on Branin–Currin; including other synthetic benchmarks (e.g., Park, Levy, Rosenbrock)
and real-world problems (e.g., Mechanical Plate Vibration Design, Thermal Conductor Design, NAS) would strengthen the evidence.
4. The paper lacks comparisons to recent joint forward–inverse operator methods such as Latent Neural Operator (LNO) [5].
I also recommend evaluating the FUSE pipeline on additional PDE tasks beyond Darcy flow (for example airfoil, Navier-Stokes)

The paper lacks empirical comparisons with crucial baselines that could be straightforwardly adjusted to the proposed setting.
This omission weakens the overall experimental support for the claimed effectiveness of the method.

**Questions:**

1. Which specific OCL strategy was implemented or was it simply a replay buffer? What motivated this choice?
2. Beyond computational savings, does the use of OCL provide any additional benefit over
simply retraining the surrogate model with concatenated data in an online optimization setting? If so, can you explain it
3. Could you please provide an analysis showing how sensitive your results are to the choice of $\lambda$ (the weighting factor for the
sensitivity-constraint term), and explain how this parameter should be selected for each task?

**Missing related works** :

[1] Repeated Augmented Rehearsal: A Simple but Strong Baseline for Online Continual Learning

[2] Online Curvature-Aware Replay: Leveraging 2nd Order Information for Online Continual Learning

[3] Multi‑Fidelity Multi‑Objective Bayesian Optimization: An Output Space Entropy Search Approach

[4] Information‑Theoretic Multi‑Objective Bayesian Optimization with Continuous Approximations

[5] Latent Neural Operator for Solving Forward and Inverse PDE Problems

[6] Holistic Physics Solver: Learning PDEs in a Unified Spectral-Physical Space

[7] Parameterized Physics-informed Neural Networks for Parameterized PDEs

---

> ### Author Response · Authors · 2025-11-26
>
> We are grateful for the reviewers’ careful and constructive comments on our submission. However, we note that the numerical scores appear more critical than the written assessments might suggest — which were largely encouraging and requested reasonable refinements. While we would address aforementioned concerns in a revised manuscript, we wished to first note this potential misalignment for the reviewers consideration.
>
> 1. Which specific OCL strategy was implemented or was it simply a replay buffer? What motivated this choice?
>
> 	Thanks for pointing this out. Our OCL strategy is based on a replay buffer following these steps: (1) create an initial buffer and use it to pretrain the surrogate at initialization; (2) during parameter inversion, randomly select and update a minibatch in the buffer at each training iteration. This new minibatch uses parameter estimates generated by the current inversion module and corresponding physical model outputs as target data. What motivated us to use this strategy is both efficiency and model stability. As the parameter space evolves dynamically during optimization, it is important for the surrogate to be exposed to the newly exploited regions of the parameter space. The replay buffer provides this balance by maintaining a mixed representation of older and newly generated samples.
>
> We chose this as a simple implementation in order to focus on the core interaction between OCL and SC. However, as the reviewer alludes to, the nuance of what OCL implementation is best in these modeling settings is an open question. We will address this decision and intended directions for continued work in revisions.
>
> 2. Beyond computational savings, does the use of OCL provide any additional benefit over simply retraining the surrogate model with concatenated data in an online optimization setting? If so, can you explain it
>
> Thanks for the opportunity to clarify. In fact, simply retraining was the initial path we attempted, but the buffer can bloat in size very quickly as the optimization proceeds (as it runs through thousands of minibatches) and the retraining from scratch can gradually take longer and longer until it slows down the entire workflow. As the reviewer noted, OCL is a substantially more efficient path to updating the surrogate model as parameters evolve, avoiding the high cost of repeatedly retraining the surrogate from scratch. OCL functions as an evolutionary update strategy: it establishes a strong initial representation and then incrementally fine-tunes the surrogate at each iteration using newly generated data. This incremental approach provides several advantages beyond computational savings: (1) higher stability by avoiding full model resets which can disrupt learning during parameter inversion; (2) better conditioning by using the surrogate’s previous knowledge as a prior during optimization; (3) continuity by avoiding using a newly retrained surrogate each iteration; (4) smoother updates by allowing smoother changes in the surrogate, and thus smoother parameter updates; (5) tighter coupling with the parameter inversion loop which helps with respect to computational efficiency.
> This said, we think the main focus here is on the interaction and combined effect of OCL and SC. We admit there are much more sophisticated strategies for OCL which we would explore further..
>
> 3. Could you please provide an analysis showing how sensitive your results are to the choice of  (the weighting factor for the sensitivity-constraint term), and explain how this parameter should be selected for each task?
>
> For all experiments presented in this paper, we have employed an automatic loss weighting scheme that learns a trainable parameter λ to automatically balance the data-fitting loss (Ldata) and the sensitivity-constrained (LSC) term during surrogate model training. The only exception is the FUSE experiment, where λ was set to 1 for simplicity. In all other cases, λ is fully trainable and adapts automatically.

---

### Official Review · Reviewer_MJ5d · 2025-11-01

**Soundness:** 2
**Presentation:** 3
**Contribution:** 2
**Rating:** 2
**Confidence:** 3

**Summary:**

This paper proposes combining Online Continual Learning (OCL) with Sensitivity Constraints (SC) to improve surrogate model performance in optimization tasks involving expensive numerical simulations. The core methodology augments standard data fitting loss with a sensitivity-matching term that aligns the Jacobian of the surrogate model with that of the physical model. The authors evaluate this approach across three optimization frameworks: (1) multi-objective multi-fidelity Bayesian optimization using the Branin-Currin benchmark, (2) hybrid end-to-end training of neural networks coupled with differentiable hydrological (dHBV) and ecosystem (δpsn) models, and (3) the FUSE framework for joint generative parameter inversion and surrogate training on 2D Darcy flow. Results show that OCL+SC consistently outperforms offline surrogates, OCL-only, and offline+SC baselines, with R2 improvements from 0.45 (offline) to 0.71 (OCL+SC) in the hydrological case and from 0.21 to 0.45 in the ecosystem case.

**Strengths:**

The evaluation across three structurally different optimization frameworks (MOMF-BO, hybrid differentiable training, joint generative-forward modeling) provides evidence of generality beyond a single application context. Each framework uses surrogates differently, yet OCL+SC consistently improves performance.

The failure of accurate surrogate models to provide good optimization guidance is a genuine challenge in computational science and engineering. The paper correctly identifies that sensitivity fidelity is often overlooked in surrogate model evaluation.

The hydrological experiments use the widely-recognized CAMELS dataset (531 catchments), and the ecosystem experiments use SAPFLUXNET (100+ sites, 120+ species), lending credibility to the validation on real observational data rather than only synthetic problems.

**Weaknesses:**

The paper extensively discusses SC-FNO (Behroozi et al., 2024), DINO (O'Leary-Roseberry et al., 2024), and DE-DeepONet (Qiu et al., 2024) but does not include them as baselines. The "Offline+SC" baseline appears to be a simple implementation rather than these published methods. The central claim that OCL+SC outperforms existing approaches cannot be validated without direct comparison. Specifically, the paper does not test whether applying online continual learning to SC-FNO or DINO would yield similar results, which is the most direct competing approach.

With only 5 random seeds and no hypothesis testing, the statistical validity of the results is questionable. The ecosystem model result where OCL+SC surpasses the physical model benchmark (0.45 > 0.438) is particularly concerning and receives no investigation. Effect sizes, p-values, and confidence intervals are entirely absent, making it impossible to assess whether observed differences are meaningful or due to chance.

The FUSE experiment uses 1000 samples on a 32×32 grid, which is orders of magnitude smaller than typical neural operator papers in 2024-2025 (often 10,000+ samples on 256×256+ grids). The Branin-Currin benchmark is a 2D analytical function, not a physical simulation, contradicting the paper's emphasis on "expensive physical models" and "high-dimensional problems." The experimental scale does not match the claimed scope.

Table 5 claims 1/30 speedup but the comparison is unfair. The physical model's 4176s/epoch likely includes adjoint gradient computation, which the authors also use for SC. The analysis omits initial buffer generation cost, cumulative OCL evaluation costs, and end-to-end time-to-convergence. A complete cost accounting would likely show much smaller speedups. The claim should compare total cost to achieve equivalent performance, not per-epoch runtime for different operations.

Key design decisions lack justification: (a) Why constrain only the "four most sensitive parameters" in the hydrology experiment rather than all 12? (b) Why evaluate gradients at only "middle and last timesteps" rather than all 730 timesteps? (c) How sensitive are results to λ (set to 1 in FUSE, unspecified elsewhere)? (d) How does the number of SC sampling points M affect results? (e) What sampling strategy for SC evaluation points is best (random, uniform, adaptive)? Without these ablations, it is unclear when and how to apply the method.

**Questions:**

Why are SC-FNO, DINO, and DE-DeepONet not included as baselines? These are the most directly relevant competing methods. How does OCL+SC compare to simply applying online continual learning to SC-FNO? Without this comparison, the contribution cannot be validated. If computational constraints prevented full comparison, can you at least test SC-FNO on one domain?

Please provide statistical significance tests. For all results in Table 1, report p-values (paired t-tests or Wilcoxon signed-rank tests) comparing OCL+SC to baselines. With standard deviations of 0.02-0.07 and only 5 seeds, are the observed differences statistically significant at p<0.05?

How does the ecosystem model surrogate outperform the physical model benchmark (R2: 0.45 vs 0.438)? This is physically implausible. Is this: (a) overfitting to the validation set, (b) an issue with the physical model implementation, (c) noise in observations, or (d) statistical fluctuation? Please investigate and explain.

Please provide complete computational cost accounting. For the ecosystem model, report: (a) cost to generate initial buffer, (b) cumulative cost of all physical model evaluations during OCL, (c) gradient computation cost for SC, (d) end-to-end time to reach R²=0.42 for each method. What is the true total speedup when all costs are included?

What accounts for the inconsistency where OCL alone helps in hydrology but hurts in FUSE? Under what conditions does OCL improve vs. degrade performance? This inconsistency suggests important boundary conditions that should be characterized.

---

> ### Author Response · Authors · 2025-11-26
>
> We are grateful for the reviewers’ careful and constructive comments on our submission. However, we note that the numerical scores appear more critical than the written assessments might suggest — which were largely encouraging and requested reasonable refinements. While we would address aforementioned concerns in a revised manuscript, we wished to first note this potential misalignment for the reviewers consideration.
>
> 1. Why are SC-FNO, DINO, and DE-DeepONet not included as baselines? These are the most directly relevant competing methods. How does OCL+SC compare to simply applying online continual learning to SC-FNO? Without this comparison, the contribution cannot be validated. If computational constraints prevented full comparison, can you at least test SC-FNO on one domain?
>
> Thank you for this suggestion! We would like to clarify that we did implement SC-FNO as baseline for the second experiment presented in the paper “HYBRID TRAINING OF NNS COUPLED TO PROCESS-BASED EQUATIONS”. This baseline corresponds to the offline + SC case in Table 1. In this setup, we pretrained an SC-FNO surrogate for the ecosystem model (case B) and an LSTM surrogate for the hydrological model (case A). For case A, the pretrained surrogate reached a maximum of R2 of 0.65 compared to 0.72 obtained by the physical  model, while for case B, the model reached a maximum  R2 of 0.35 compared to 0.438 for the physical model. These results illustrate the limitations of only pretraining offline SC-FNO and highlights the benefits of OCL + SC, where the surrogate continues to be fine-tuned during parameter inversion rather than being trained only once on the initial buffer.
>
> 2. Please provide statistical significance tests. For all results in Table 1, report p-values (paired t-tests or Wilcoxon signed-rank tests) comparing OCL+SC to baselines. With standard deviations of 0.02-0.07 and only 5 seeds, are the observed differences statistically significant at p<0.05?
>
> Thank you for the suggestion! We have implemented paired-t-tests on the R2 results in Table 1,  comparing OCL + SC to baseline surrogates (considered in this case as offline, OCL, offline + SC).
>  - Using offline as the baseline case: all other cases yield p-values < 0.05 indicating significant improvement in results (surrogate fidelity) when using online continual learning alone (OCL, second row), or SC alone (offline + SC, third row), or their combination (OCL + SC, from fourth row till end).
>
> - Using OCL or offline+SC as baselines, all variants for the ecosystem model remain statistically significant (p < 0.05) except OCL + SC (1) and OCL + SC (10). This is similar for the hydrological model except when OCL is used as the baseline, where all variants are statistically significant.
>
> These results show that introducing OCL, SC, or OCL + SC significantly improves performance over the purely offline surrogate (pre-training only). The lack of significance for OCL + SC (1) and OCL + SC (10) relative to OCL or Offline + SC may be attributed to high frequent fine-tuning in OCL + SC (1) leading to overfitting, while infrequent fine-tuning in OCL+ SC (10) may be under-fitting.
>
> | Model / Method  | Offline | OCL   | Offline + SC |
> |-----------------|---------|-------|---------------|
> | **Hydrologic Model (p-value for R²)** |       |       |               |
> | Offline         |   -     | **0.024** | **0.024**     |
> | OCL             | **0.024**   |   -   | 0.081         |
> | Offline + SC    | **0.010**   | 0.081 |   -           |
> | OCL + SC (1)    | **0.015**   | **0.003** | 0.057         |
> | OCL + SC (2)    | **0.013**   | **0.004** | **0.020**     |
> | OCL + SC (5)    | **0.011**   | **0.001** | 0.053         |
> | OCL + SC (10)   | **0.034**   | **0.007** | 0.238         |
> | **Ecosystem Model (p-value for R²)** |       |       |               |
> | Offline         |   -     | **0.049** | **0.014**     |
> | OCL             | **0.049**   |   -   | **0.028**     |
> | Offline + SC    | **0.014**   | **0.028** |   -           |
> | OCL + SC (1)    | **0.027**   | 0.050 | 0.561         |
> | OCL + SC (2)    | **0.008**   | **0.002** | **0.015**     |
> | OCL + SC (5)    | **0.009**   | **0.005** | **0.045**     |
> | OCL + SC (10)   | **0.049**   | 0.302 | 0.479         |

---

> > ### Author Response · Authors · 2025-11-26
> >
> > 3. How does the ecosystem model surrogate outperform the physical model benchmark (R2: 0.45 vs 0.438)? This is physically implausible. Is this: (a) overfitting to the validation set, (b) an issue with the physical model implementation, (c) noise in observations, or (d) statistical fluctuation? Please investigate and explain.
> >
> > We would like to clarify that the reported R2 for the physical model comes from a single run with a fixed random seed, whereas the surrogate performance is averaged over 5 random seeds. This makes it possible for the surrogate to slightly exceed the single-seed physical benchmark due to statistical fluctuations. Additionally, the physical ecosystem model can experience numerical instabilities under very dry conditions which may negatively impact the parameter optimization, whereas the surrogate, on the other hand, provides more stable and smoother predictions under similar conditions.
> > So we interpret these differences as a combination of (b) and (d), we have also explored multi-seed evaluations of the physical model itself to provide a more robust benchmark. Results in Table 2 were updated as the following:
> >
> > | Model                     | R2_Hydrologic      | Bias_Hydrologic     | R2_Ecosystem      | Bias_Ecosystem      |
> > |---------------------------|---------------------|----------------------|-------------------|----------------------|
> > | Physical Model (Benchmark)| **0.72 ± 0.01**     | **0.602 ± 0.02**     | **0.46 ± 0.01**   | **-0.011 ± 0.005**   |
> > | Offline                   | 0.45 ± 0.07         | 0.70 ± 0.05          | 0.21 ± 0.12       | -0.059 ± 0.012       |
> > | OCL                       | 0.63 ± 0.03         | 0.69 ± 0.05          | 0.41 ± 0.02       | -0.022 ± 0.005       |
> > | Offline + SC              | 0.65 ± 0.05         | 0.69 ± 0.05          | 0.35 ± 0.04       | -0.030 ± 0.015       |
> > | OCL + SC (1)              | 0.69 ± 0.02         | 0.64 ± 0.02          | 0.42 ± 0.04       | -0.019 ± 0.005       |
> > | OCL + SC (2)              | **0.71 ± 0.02**     | **0.62 ± 0.02**      | **0.45 ± 0.02**   | **-0.014 ± 0.003**   |
> > | OCL + SC (5)              | 0.70 ± 0.03         | 0.63 ± 0.02          | 0.43 ± 0.02       | -0.017 ± 0.003       |
> > | OCL + SC (10)             | 0.68 ± 0.04         | 0.67 ± 0.03          | 0.39 ± 0.06       | -0.023 ± 0.005       |
> >
> > 4. Please provide complete computational cost accounting. For the ecosystem model, report: (a) cost to generate initial buffer, (b) cumulative cost of all physical model evaluations during OCL, (c) gradient computation cost for SC, (d) end-to-end time to reach R²=0.42 for each method. What is the true total speedup when all costs are included?
> >
> > We would like to clarify that the timings reported in Table 5 (in seconds) represent the full simulation time divided by the total number of epochs. Aside from the cost to generate the initial buffer, it includes (b) the cumulative cost of all physical model evaluations during OCL, (c) gradient computation cost for SC, and the total cost required to reach the final performance reported in Table 1 after 50 epochs. So We have now added the cost of creating the initial buffer and report the complete simulation time for each run in the table below.
> >
> > | Model                     | Total Run Time (seconds) | % of Time Relative to Benchmark | Run Time per Epoch (seconds) |
> > |---------------------------|---------------------------|----------------------------------|-------------------------------|
> > | Physical Model (Benchmark)| 193100                    | 100                              | 3862                          |
> > | Offline                   | 4450                      | 2.30                             | 17                            |
> > | OCL                       | 10750                     | 5.57                             | 143                           |
> > | Offline + SC              | 4650                      | 2.41                             | 21                            |
> > | OCL + SC (1)              | 12600                     | 6.53                             | 180                           |
> > | OCL + SC (2)              | 10800                     | 5.59                             | 144                           |
> > | OCL + SC (5)              | 9600                      | 4.97                             | 120                           |
> > | OCL + SC (10)             | 8950                      | 4.63                             | 107                           |

---

> > > ### Author Response · Authors · 2025-11-26
> > >
> > > 5. What accounts for the inconsistency where OCL alone helps in hydrology but hurts in FUSE? Under what conditions does OCL improve vs. degrade performance? This inconsistency suggests important boundary conditions that should be characterized.
> > >
> > > There may be a misunderstanding here: OCL did not hurt performance in the FUSE experiment. Rather, we believe the baseline surrogate in FUSE was already sufficiently accurate such that the incremental benefit of OCL was smaller than in the hydrology case. Indeed, compared with the hydrological model, the parameter space for FUSE was lower dimensional and so the surrogate maintained accuracy even without continual updates making the relative gain from OCL less pronounced.
> > > Generally, OCL yields the largest benefits when the optimization explores a wide and non-stationary parameter space where a surrogate may be prone to concept drift or local overfitting. Continual refinement helps to track with these changes. Conversely, if the surrogate is already situated in a stable and relevant region of parameter space, OCL may only provide moderate, not detrimental, improvement. We will clarify these conditions in the manuscript.

---

### Official Review · Reviewer_jTub · 2025-11-01

**Soundness:** 3
**Presentation:** 2
**Contribution:** 2
**Rating:** 6
**Confidence:** 3

**Summary:**

This paper proposes combining online continual learning (OCL) with scheduled sensitivity constraints (SC) to improve surrogate-based optimization. The idea is to continually refine a surrogate model with incremental data while periodically enforcing consistency between surrogate sensitivities (Jacobians) and those of the true numerical model. The authors argue that this hybrid approach better preserves meaningful gradients for long-horizon, high-dimensional optimization tasks. The method is evaluated across several settings including multi-fidelity Bayesian optimization, hybrid physics–ML models, and generative parameter inversion frameworks.

The paper tackles an important problem with broad applicability, namely improving surrogate fidelity, particularly with respect to sensitivities, to support efficient optimization. The results generally support the claim that OCL combined with SC improves performance over OCL or SC alone. However, the contribution would benefit from clearer positioning relative to extensive prior work in active learning, adaptive surrogate refinement, and dynamic model updating, as well as more clarity on the application context and practical meaning of improvements. While promising, the novelty and scope currently feel somewhat diffuse due to the breadth of examples and insufficient literature contextualization.

**Strengths:**

Originality: The paper addresses sensitivity preservation in surrogate modeling, a recognized challenge in surrogate-based optimization. Novel combination of continual learning and sensitivity constraints, and the scheduling strategy for SC injection is interesting. Diverse illustrative applications across optimization, hybrid modeling, and inverse problems show potential generality.

Quality: Empirical results generally support claims that OCL+SC yields better optimization trajectories than alternatives and the paper demonstrates that sparse or infrequent SC finetuning can still give meaningful gains.

Clarity: The motivation is clear, and algorithm steps are reasonably described. Claims are consistently stated and supported numerically. It is really positive that the authors based their work on available open-source codes that are acknowledged and discuss the generalizability of this approach as differentiable programming becomes more available (which the Reviewer agrees is true).

Significance: If properly contextualized, the idea could be useful in large-scale scientific surrogate-assisted optimization. The observation that limited initial data can suffice (if demonstrated rigorously) could be impactful for expensive simulations.

**Weaknesses:**

1. The manuscript engages in an active research area (surrogate-assisted learning, continual learning, active/adaptive data acquisition) but does not sufficiently clarify: How OCL relates to or differs from adaptive learning, online learning, or active learning in scientific modeling Whether prior works combining sensitivities with incremental data exist in related communities (e.g., multifidelity active learning, physics-guided update strategies)? Without clearer boundaries, the novelty claim (first to combine SC with OCL) is difficult to verify. Actionable suggestion: explicitly define continual learning vs. adaptive surrogate refinement vs. online active learning, and cite key lines of work in each.  In literature review, explain examples from a clearly continual learning need (e.g., not general field of "design using surrogates", which is a huge area where is continual learning needed, or not?)

2. Related to above comment, it is not clear whether the method targets: Dynamic systems with evolving parameters, or static systems where data is progressively acquired to improve surrogates, or both? The mixed examples blur this distinction and make it harder to map contributions to existing streams. Clarify target problem class and restructure literature and examples accordingly.

3. Practical significance not well discussed: While results improve metrics, the real impact is unclear. For instance: In the hydrological case, what does a validation error difference of 1.3 vs 1 translate to physically? Does the sensitivity improvement significantly change real-world decisions? Suggestion: add brief discussion connecting numerical improvements to domain relevance.

4. Experimental clarity: Dimensionality and complexity of the hydrological model are insufficiently specified. Results focus on relative loss improvement; little insight into uncertainty, stability, or robustness. Suggestion: provide dimensionality details, and if possible discuss sensitivity to noise/initialization.

5. Breadth vs depth: Multiple different applications are showcased, which suggests generality, but also makes evaluation feel surface-level. Suggestion: Consider deepening analysis in one domain or adding a conceptual unifying framework to help the reader navigate the diversity of settings.

**Questions:**

1. How exactly do you define “continual learning” in this context, and how does it substantively differ from well-established active learning, adaptive surrogate modeling, or online Bayesian optimization?

2. Is the approach primarily intended for dynamic systems that evolve over time, or static systems where additional samples are sequentially acquired, or both? Specifically in case of static systems, what are some practical examples of a steady-state design problem that would need continual learning vs. active learning for example?

3. Can you detail the dimensionality and computational scale of the hydrological test case? How challenging is it relative to existing hydrology benchmarks?

4. What does the improvement in surrogate loss translate to in terms of real decision-making or physical interpretability in the hydrological and other settings?

5. Did you consider uncertainty-aware baselines (e.g., BO with uncertainty-driven refinement)? If so, how does the method compare?

6. Is SC applicable if the physical model does not support efficient Jacobian computation, and what are scalability limits?

---

> ### Author Response · Authors · 2025-11-26
>
> We are grateful for the reviewers’ careful and constructive comments on our submission.
>
> Questions:
> 1. Question1: How exactly do you define “continual learning” in this context, and how does it substantively differ from well-established active learning, adaptive surrogate modeling, or online Bayesian optimization?
>
>  In our work, we use OCL to ensure that the surrogate model maintains fidelity as the parameter space evolves during optimization. This implementation first involves pretraining the surrogate on an initial buffer (the same that would be used for offline-only training). Then, at each epoch of HV-KG/FUSE/hydrological/ecosystem training, samples in the buffer are selected at random and replaced with new ones generated by the physical model using the most recent parameter estimates from the inversion module. At each iteration, the surrogate is fine-tuned on this buffer, not retrained from scratch, thereby allowing it to retain previously learned structure while adapting to newly explored regions of the state space. This method differs from the three approaches mentioned by the reviewer:
>   - Active learning typically selects informative points from a fixed pool of unlabeled data, and labels them, but does not generate new labeled samples as in our setting.
>   - Adaptive surrogate modeling often rebuilds or retrains the surrogate from scratch every iteration whenever new samples are added, rather than incrementally updating it with prior knowledge as we do, so it does not constitute continual training.
>   - Online Bayesian optimization shares the idea of sequential updates, but is restricted to probabilistic surrogates, whereas continual learning applies broadly to a variety of surrogate architectures beyond the Bayesian setting.
>
> 2.  Question 2: Is the approach primarily intended for dynamic systems that evolve over time, or static systems where additional samples are sequentially acquired, or both? Specifically in the case of static systems, what are some practical examples of a steady-state design problem that would need continual learning vs. active learning for example?
>
>  OCL is applicable to both static and dynamic systems. In particular, OCL does not address drift of parameters in physical time, but rather intends to correct for their evolution across a parameter space during optimization. In our hydrological and ecosystem experiments, continual learning is beneficial because the surrogate is taking parameter inputs from a NN that is training and evolving its parameter space from those initially seen in the surrogate’s buffer. That is, as the loss propagates, the optimizer moves through different regions of the parameter space to reduce the mismatch between predictions and observations, and the surrogate must adapt accordingly.
>
>  This is a similar story for the multi-objective multi-fidelity HV-KG example. While the objectives are physical time-invariant, the parameter space evolves over the course of optimization and deviates from the original samples the surrogate was trained on.
> Even in steady-state settings such as the FUSE Darcy example, the parameters vary across the spatial domain, and continual learning helps the surrogate remain accurate as different parts of the parameter space are explored during training. It can also be beneficial in terms of achieving optimized engineering designs and inversion.
>
> 3. Question 3: Can you detail the dimensionality and computational scale of the hydrological test case? How challenging is it relative to existing hydrology benchmarks?
>
>  Hybrid model learning as in the hydrological test presents a high-dimensional and computationally intensive learning task. The intention in this physics-ML coupled setting is to train a parameterization NN that takes 39 inputs and outputs 194 parameters for the physical hydrological model. Both the NN and hydrological models operate over a recurrent sequence of 730 daily time steps, which substantially increases the effective dimensionality of the learning task and the cost of gradient-based optimization. Combined, high-dimensional parameterization, long recurrent dependencies, non-linear process-based simulation within the physical model, and concurrent optimization for 531 sample locations makes this task complex even with the advantage of GPU parallelization. Furthermore, as the physical model (HBV) used here is relatively simple compared with other contemporary hydrological models, scale effects mean increasing physical complexity will generally amplify the computational burden of this optimization making surrogate models more relevant as hybrid modeling accelerators.

---

> > ### Author Response · Authors · 2025-11-26
> >
> > 4. Question 4: What does the improvement in surrogate loss translate to in terms of real decision-making or physical interpretability in the hydrological and other settings?
> >
> >  Improvements in surrogate loss translate directly into higher fidelity between the surrogate and underlying physical model. In practice, this should mean that the surrogate model’s predictions – and, therefore, the gradients used during optimization – more closely mirror those produced by the full hydrological or ecosystem model. When the surrogate is accurate, the optimization trajectory driven by it in the hydrological/ecosystem settings converges toward the same parameter estimates and model states that would be obtained if we had used the physical model itself at every step. Without this guarantee, the outcome of the optimization is useless. Without the OCL+SC procedures outlined here, we can often find critical divergence in learning outcomes between using the surrogate and physical model. We will add discussion here.
> > What is important, is that we can now finish a learning task, e.g., the ecosystem model which used to take many days, to mere hours. This critically enables the testing of many trajectories and alternative model structures. We hope to share this experience with those communities who now want to achieve coupled hybrid learning of physical models and neural networks.
> >
> > 5. Question 5: Did you consider uncertainty-aware baselines (e.g., BO with uncertainty-driven refinement)? If so, how does the method compare?
> >
> >  The surrogate framework proposed here can be seen as complementary to uncertainty-aware baselines such as the BO with uncertainty-driven refinement that the review mentions. The present work does not implement a full Bayesian treatment or uncertainty-propagating surrogate. However, our surrogates will critically enable these methods. The methods we explained in this paper, e.g., MO-MFBO and FUSE, naturally lend themselves to Bayesian uncertainty quantification. While the hybrid modeling is itself not uncertainty-aware, it provides the computational efficiency required to run ensemble-based uncertainty quantification or uncertainty-driven refinement at scale. The incorporation of explicit uncertainty modeling is a natural next step, and we will clarify this point in the revised manuscript.
> >
> >
> > 6. Question 6: Is SC applicable if the physical model does not support efficient Jacobian computation, and what are scalability limits?
> >
> >  Thanks for asking. In fact, if a model can be implemented on a modern ML platform, then it already supports gradients through automatic differentiation. If not, sensitivities can still be obtained through standard approaches such as adjoint methods, which have been common in many computational domains, or through finite-difference approximations when adjoints are impractical. Low-rank techniques also provide efficient alternatives when the full Jacobian is too costly.
> >
> >  We also note that applying SC does not require computing the full Jacobian. For example, in our experiments, using randomly sampled sensitivities was sufficient to demonstrate key benefits. This flexibility, therefore, means SC can be applied even when full derivative information is unavailable or expensive to compute.

---

### Note · Authors · 2025-12-03

**Comment:**

We appreciate the reviewers’ constructive feedback. We noted that the numerical scores were more critical than the written assessments, which were largely positive and requested reasonable refinements. Before the ICLR leak, we had planned to flag this misalignment for consideration. We also aim to add more case studies to show the tool’s practical value, so a withdrawal and revision is the most appropriate next step.

**Withdrawal Confirmation:**

I have read and agree with the venue's withdrawal policy on behalf of myself and my co-authors.